# Multilingual Anchoring: Interactive Topic Modeling and Alignment Across Languages

**Michelle Yuan**
University of Maryland
myuan@cs.umd.edu

**Benjamin Van Durme**
John Hopkins University
vandurme@jhu.edu

**Jordan Boyd-Graber**
University of Maryland
jbg@umiacs.umd.edu

## Abstract

Multilingual topic models can reveal patterns in cross-lingual document collections. However, existing models lack speed and interactivity, which prevents adoption in everyday corpora exploration or quick moving situations (e.g., natural disasters, political instability). First, we propose a multilingual anchoring algorithm that builds an anchor-based topic model for documents in different languages. Then, we incorporate interactivity to develop MTAnchor (Multilingual Topic Anchors), a system that allows users to refine the topic model. We test our algorithms on labeled English, Chinese, and Sinhalese documents. Within minutes, our methods can produce interpretable topics that are useful for specific classification tasks.

## 1   Introduction: Exploring multilingual document collections

Modeling multilingual topics aids exploration of large corpora across languages [1]. These models help align topics cross-lingually and uncover latent relationships between languages, such as observing the differences in describing economic issues between English and Spanish speakers [2]. Incorporating multilingual information also forms better monolingual topics [3].

Multilingual topic models usually depend on some resource to bridge languages. These resources include word alignments [4], dictionaries [3, 5], topic alignments in documents [6], or all of the above [7]. Existing multilingual models have several shortcomings; they assume extensive knowledge about languages, preclude human refinement, and are slow. Thus, a topic model may not be appropriate in emergent sitations on low resource languages when time is of the essence: e.g., when relief workers must triage relief messages in Hatian Creole [8].

Beyond these practical concerns, adding interactivity to topic modeling allows machine learning non-experts to build models better suited to their needs [9–11]. One way to quickly incorporate human knowledge into the model is through anchor words [12]. Inference in anchor-based topic models is driven by anchors, which are words that have high probability in one topic and low probability in remaining topics [13, 14]. The anchoring algorithm scales with the number of unique word types, making it fast enough for interactive updates.

We present two contributions for modeling multilingual topics. First, we develop a multilingual anchoring algorithm, which is an extension to anchor-based topic inference for comparable corpora.[1] Second, we introduce MTAnchor, a human-in-the-loop system that uses multilingual anchoring to align topics and enables users to make further adjustments to the model.[2] Through interaction, the model produces *interpretable*, low-dimensional representations of documents. These vector representations improve intra-lingual or cross-lingual text classification. The topic model generates coherent topic aligments for comparable corpora because users themselves align topics.

## 2 Anchor-based topic models

A topic model discovers topics: a distribution over words that evinces a coherent theme [17]. Well-known methods for constructing topic models are latent Dirichlet allocation [18, LDA] and latent semantic analysis [19, LSA]. Another computationally attractive option is the anchor word algorithm [13] that uses the row-normalized word co-occurrence matrix $\bar{Q}$, where $\bar{Q}_{i,j} = p(w_2 = j \mid w_1 = i)$. The vector $\bar{Q}_i$ is the $i^{th}$ row of $\bar{Q}$ and represents the conditional distribution of words in a document given that word $i$ has occurred. Anchor word $s$ appears with high probability in only one topic, so $\bar{Q}_s$ resembles a topic's word distribution in topic models like LDA. For example, if "concealer" is an anchor word for a cosmetics topic, then its conditional distribution will have high probability for cosmetics-related words and low probability for other words. Still, these are not the distributions that typically define probabilistic topic models: the probability of a word given a topic.

### 2.1 Anchoring

To discover topic distributions, anchor word approaches [14] search for coefficients that describe non-anchor words' document contexts with anchor words' conditional distributions. The word "liner" has meanings that are explained by "album" in a music topic, "concealer" in a cosmetics topic, and "carburetor" in an automotive topic. Then, the conditional distribution of "liner" can be expressed as a convex combination of the conditional distributions of "album", "concealer", and "carburetor". Given anchor words $s_1, \ldots, s_K$, the conditional distribution of word $i$ can be approximated as

$$\bar{Q}_i \approx \sum_{k=1}^{K} C_{i,k} \bar{Q}_{s_k} \quad \text{subject to} \sum_{k=1}^{K} C_{i,k} = 1 \text{ and } C_{i,k} \geq 0. \tag{1}$$

The coefficient $C_{i,k}$ represents $p(z = k \mid w = i)$, the probability of topic $k$ given a word $i$. These coefficients are recovered using the RecoverL2 algorithm [14], which minimizes the quadratic loss between $\bar{Q}_i$ and $\sum_{k=1}^{K} C_{i,k} \bar{Q}_{s_k}$. Using Bayes' rule, we can obtain the standard topic matrix $A$,

$$A_{i,k} = p(w = i \mid z = k) \propto p(z = k \mid w = i)p(w = i) = C_{i,k} \sum_{j=1}^{V} \bar{Q}_{i,j}. \tag{2}$$

For a large vocabulary size $V$, finding these anchor words is a challenge, but understanding the geometric intuition behind the anchoring algorithm can help us select the right words. Points inside a convex hull are expressed as the convex combination of their vertices. If we want to approximate $\bar{Q}_i$ as the convex combination of $\bar{Q}_{s_1}, \ldots, \bar{Q}_{s_K}$ (Equation 1), then $\bar{Q}_{s_1}, \ldots, \bar{Q}_{s_K}$ should be the vertices of the convex hull of $\bar{Q}$. However, finding the vertices to a $V$-dimensional convex hull is time-consuming [13]. Instead, Arora et al. [14] use FastAnchorWords, a greedy approach similar to Gram-Schmidt orthogonalization, to construct an approximate convex hull of $\bar{Q}$ and expand it as much as possible with each choice of anchor word. Other methods include projecting $\bar{Q}$ to a low-dimensional space and finding the vertices of its exact convex hull [20], adding another dimension to capture metadata [21], or finding nonparametric anchor words [22].

### 2.2 Multiword anchoring

Finding topics in anchor-based models is fast, so it can be used in an interactive setting where users iteratively choose anchor words for every topic [12]. Nevertheless, users may want to choose multiple anchor words for a topic, such as selecting both "concealer" and "lipstick" for a cosmetics topic. Therefore, Lund et al. [12] propose multiword anchoring: users select a set $\mathcal{G}_k$ of multiple anchor words for topic $k$. After users select $\mathcal{G}_1, \ldots, \mathcal{G}_K$, $\bar{Q}$ is augmented so that new rows $\bar{Q}_{V+1}, \ldots, \bar{Q}_{V+K}$ represent these pseudo-anchors in the conditional word co-occurrence space. Lund et al. [12] construct these vectors $\bar{Q}_{V+k}$ as

$$\bar{Q}_{V+k,j} = \left( \frac{\sum\limits_{i \in \mathcal{G}_k} \bar{Q}_{i,j}^{-1}}{|\mathcal{G}_k|} \right)^{-1}. \tag{3}$$

The motivation for using the harmonic mean (Equation 3) is that the function can centralize input values and ignore large outliers. Finding topics follows the same algorithm as before using single word anchors. Instead of modeling $\bar{Q}_i$ as the convex combination of $\bar{Q}_{s_1}, \ldots, \bar{Q}_{s_K}$, a convex combination of $\bar{Q}_{V+1}, \ldots, \bar{Q}_{V+K}$ models $\bar{Q}_i$ with minimal quadratic loss.

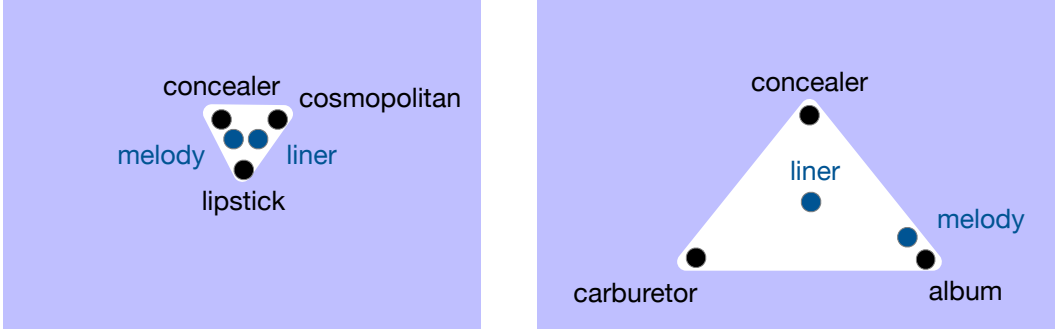

Figure 1: Visualizing the importance of choice in anchor words for approximating conditional distributions. The chosen anchor words are the black dots and their span is the white triangle. On the left, the span of anchor words is small, so the words "melody" and "liner" are too close together. On the right, the span of anchor words is large, so the conditional distributions of words "melody" and "liner" are approximated more accurately.

## 3 Bridging languages: How do you say anchor in Chinese?

Anchor-based topic models are well-defined for individual languages, but a multilingual model requires topics that are thematically connected across languages. Discovering two separate sets of anchor words does not suffice. In this section, we propose multilingual anchoring as an algorithm to cross-lingually link topics and their corresponding anchor words.

First, we can connect anchor words across languages as *anchor links*. For example, "anchor" may be linked to "锚(máo)" in Chinese under a nautical context. After anchor words are linked, all words in the same topic across languages will be form a coherent multilingual topic. A straightforward way to link words across languages is through a *dictionary*, much as a human would. Just as possessing a Chinese dictionary does not enable someone to speak Chinese, a dictionary does not magically create multilingual topics. To construct an overall coherent model, anchor links should be carefully selected.

We define these links in more detail. A language $\mathcal{L}$ is a set of word types $w$. A bilingual dictionary $\mathcal{B}$ is a subset of the Cartesian product $\mathcal{L}^{(1)} \times \mathcal{L}^{(2)}$, where $\mathcal{L}^{(1)}, \mathcal{L}^{(2)}$ are two different languages. An element $(w^{(1)}, w^{(2)})$ of $\mathcal{B}$ represents a dictionary entry where words $w^{(1)} \in \mathcal{L}^{(1)}$ and $w^{(2)} \in \mathcal{L}^{(2)}$ are translations of each other. While $\mathcal{B}$ is a binary relation, it is not necessarily a function. Other multilingual topic models require that the dictionary is a one-to-one correspondence [3, 23, 2]. We relax this restriction on $\mathcal{B}$ to extract as much information from the dictionary as possible.

We could select anchor words $s_1, ..., s_K$ *independently* for each language by considering all words $w^{(1)} \in \mathcal{L}^{(1)}$ and $w^{(2)} \in \mathcal{L}^{(2)}$ as possible candidates for anchors (e.g., independent runs of anchor algorithm). Instead, we want to *jointly* choose anchor words for both languages. First, we use dictionary entries to create *links* between words. Then, we choose anchor words $s_k^{(1)}$ for Language 1 and $s_k^{(2)}$ for Language 2 such that $s_k^{(1)}$ and $s_k^{(2)}$ are linked. Through this process, we obtain a set of $K$ anchor words for each language and can obtain topics using RecoverL2 [14].

### 3.1 Multilingual anchoring

If there is only one anchor word for each topic, our goal of building a coherent multilingual topic model would fail. Any imperfection in the dictionary would scupper the topic model. Fortunately, Arora et al. [14] assert that there exist many anchor word choices for a topic. Even if we reduce the pool for candidate anchors, we can still find suitable anchor words for each topic. Recall that anchor words are the vertices to the convex hull of words in the conditional distribution space (Section 2). Finding the actual vertices of the convex hulls is too expensive, so FastAnchorWords searches for a set of anchors with maximal span. This span should approximate the convex hull of $\bar{Q}$. Without a large enough span, we can never find accurate approximations for words in the conditional distribution space. All words $w$ will have indistinguishable conditional distributions (Figure 1). As a result, every topic will have indistinct word distributions and the resulting topics will be copies of one another.

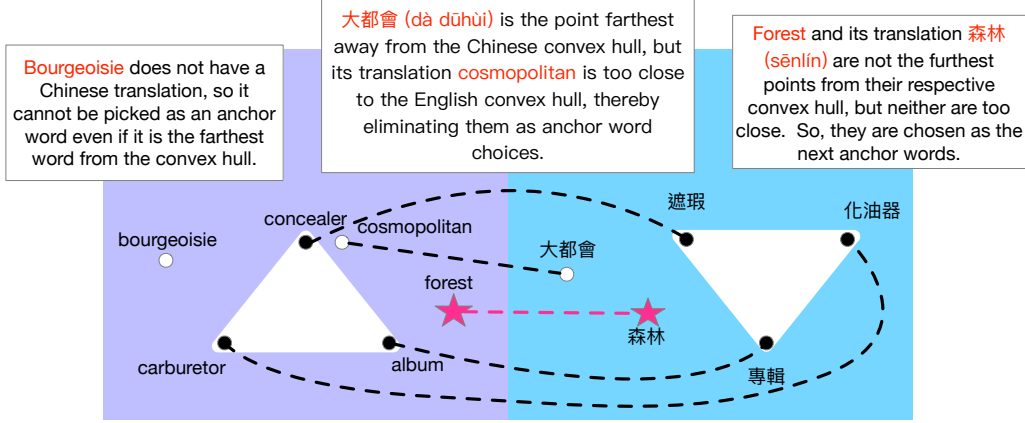

Figure 2: Selecting anchor links for multilingual anchoring. The purple (blue) area represents the conditional distribution space of words in the English (Chinese) corpus. The white triangle designates the space spanned by chosen anchor words. Dashed lines depict anchor links across spaces. Black points denote words already chosen as anchors, white points are unchosen words, and pink stars are most optimal anchors for the current iteration. Multilingual anchors should maximize area spanned by white triangles in both spaces.

To maximize span of anchor words, FastAnchorWords [14] chooses anchor word $s_k$ such that

$$s_k = \operatorname*{argmax}_{w} d\left(\operatorname{span}\left(\bar{Q}_{s_1}, ..., \bar{Q}_{s_{k-1}}\right), \bar{Q}_w\right), \tag{4}$$

where $d(P, i)$ is defined as the Euclidean distance from point $i$ to subspace $P$, or the norm of the projection of $i$ onto the orthogonal complement of $P$.

To extend the greedy approach to multilingual settings, we need anchor words that can guide topic inference in *multiple* languages. This motivates our approach for linking words with a dictionary. By choosing linked anchor words, the algorithm can align topics cross-lingually so that the aligned topics form one multilingual topic. However, randomly choosing translation pairs as anchor links will not produce coherent multilingual topics. We need multilingual anchors that also inherit the geometric properties of monolingual anchors. So, the span of anchor words should be maximized in both languages for optimal topic inference. To clearly state our objective, we define $P_j^{(l)}$ as the subspace spanned by $j$ chosen anchor words in the conditional distribution space of language $l$,

$$P_j^{(l)} = \operatorname{span}\left(\bar{Q}_{s_1^{(l)}}^{(l)}, ..., \bar{Q}_{s_j^{(l)}}^{(l)}\right). \tag{5}$$

Word $w$ is a good choice of a $k^{th}$ anchor if $\bar{Q}_w$ is far enough from $P_{k-1}^{(l)}$ so that having $\bar{Q}_w$ as an additional vertex can greatly expand span of anchors. A word might be a great choice for an anchor in one language, but we cannot select it if its translation is a poor choice for the other language (Figure 2). We need to pick linked words $w \in \mathcal{L}^{(1)}$ and $v \in \mathcal{L}^{(2)}$ such that $w$ is far from $P_{k-1}^{(1)}$ and $v$ is also far away from $P_{k-1}^{(2)}$. Then, adding $w$ and $v$ as anchor words can increase total span of anchor word set in both languages. Using this intuition, we maximize the lower bound on the distance from anchor words to $P_{k-1}^{(1)}$ and $P_{k-1}^{(2)}$. We select anchor words $w$ and $v$ such that

$$s_k^{(1)}, s_k^{(2)} = \operatorname*{argmax}_{w,v} \min\left\{ d\left(P_{k-1}^{(1)}, \bar{Q}_w^{(1)}\right), d\left(P_{k-1}^{(2)}, \bar{Q}_v^{(2)}\right)\right\} \quad \text{subject to } (w, v) \in \mathcal{B}. \tag{6}$$

We greedily select anchors $s_k^{(1)} \in \mathcal{L}^{(1)}, s_k^{(2)} \in \mathcal{L}^{(2)}$ such that Equation 6 is satisfied on every iteration $k$. Words with multiple translations are elegantly addressed: if an anchor word $w$ is picked already, then it is not likely to be picked again. The algorithm expands both convex hulls simultaneously with each iteration. Indeed, more translations aid our anchor search because there will be more linked anchors to choose from. Even if the algorithm chooses anchor words similar in meaning within the same language, interactivity can help remove duplicate topics (Section 3.2). After

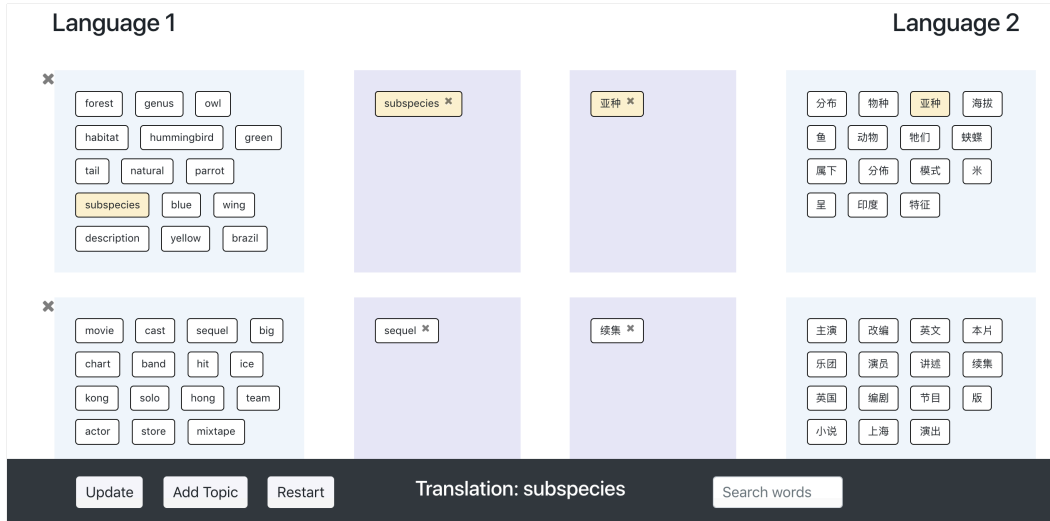

Figure 3: The user interface for exploring topics in English and Chinese documents. Anchor words are in the center, while the most likely words for each topic are on the left and right sides of the interface. The user can drag words from the side and add them as anchor words. When the user hovers over "亞種(yàzhǒng)", then its translation, "subspecies", appears at the bottom of the screen. When the user presses on the word, all occurrences of it and its translation are highlighted in yellow. Users can type words in the "Search words" box to find which words are in the vocabulary. These features help the user explore topics in an unfamiliar language.

picking a set of anchor words for each language, multilingual anchoring follows FastAnchorWords (Section 2.1). Topic matrices $A^{(1)}$ and $A^{(2)}$ are separately recovered (Equations 1, 2). These matrices are the output of multilingual anchoring. In the next sections, we show how MTAnchor further updates $A^{(1)}$ and $A^{(2)}$ based on human feedback.

**Lacking dictionary entries.** If dictionary entries are scarce, then we cannot constrain the anchor words to only be words from the dictionary. So, we independently find anchor words for each language using RecoverL2. This reduction to monolingual settings resembles other cross-lingual models: JointLDA reduces to LDA and PTLDA reduces to TLDA when there are no dictionary entries [3, 7].

**Predicting labels from topics.** Multilingual anchoring is an unsupervised method, but the topic distribution acts as a low-dimensional representation for each document [24–26]. To infer the topic distribution of documents, we pass in the topic matrices as inputs into variational inference [18], where topic variational parameter $\beta$ is fixed and only document variational parameter $\gamma$ is fitted. Then, we train a linear SVM on the topic distributions of documents [27] to classify document labels.

### 3.2 Interactive topic alignment

Multilingual anchoring uses translations to find anchor words that can lead to better topics for both languages. However, we cannot completely rely on dictionary entries to construct the topic model. In reality, translations may not be available, could be a poor fit for the dataset, or might be wrong. In addition to problems with the dictionary, the data may be too noisy, or the anchoring algorithm returns a topic model unsuited for our needs (e.g., if a user needs to separate news from opinion and the topic model puts them together). Thus, we incorporate interactivity into MTAnchor so that we can extract linguistic and cultural knowledge from humans.

First, MTAnchor takes in a comparable corpora and a bilingual dictionary as inputs. Next, it uses multilingual anchoring (Section 3.1) to find sets of anchor words for each language. After the algorithm recovers topic matrices, the interface shows information about the topic model. The user can press on the red "X" to delete any incoherent or duplicate topics (Figure 3). The user can also add new topics by pressing on "Add Topics". The interface will create a new blank row beneath the

Table 1: Comparison of multilingual topic modeling methods. Multilingual anchoring scores higher in classification accuracy and topic coherence than MCTA. MTAnchor does as well as multilingual anchoring on average, but a few users can achieve the best results for every metric.

| Dataset | Method | Classification accuracy | | | | Topic coherence | | | |
|---|---|---|---|---|---|---|---|---|---|
| | | EN-I | ZH-I SI-I | EN-C | ZH-C SI-C | EN-I | ZH-I SI-I | EN-E | ZH-E SI-E |
| Wikipedia (EN-ZH) | Multilingual anchoring | 69.49% | 71.24% | 50.37% | 47.76% | 0.141 | 0.178 | 0.084 | 0.128 |
| | MTAnchor (maximum) | **80.71**% | **75.33**% | **57.62**% | **54.54**% | **0.195** | **0.198** | **0.103** | **0.147** |
| | MTAnchor (median) | 69.49% | 71.44% | 50.27% | 47.22% | 0.141 | 0.178 | 0.084 | 0.129 |
| | MCTA | 51.56% | 33.35% | 23.24% | 39.79% | 0.126 | 0.085 | 0.000 | 0.037 |
| Amazon (EN-ZH) | Multilingual anchoring | **59.79**% | **61.10**% | **51.73**% | **53.20**% | **0.069** | **0.061** | **0.031** | **0.045** |
| | MCTA | 49.53% | 50.64% | 50.27% | 49.49% | -0.028 | 0.019 | 0.017 | 0.011 |
| LORELEI (EN-SI) | Multilingual anchoring | **20.78**% | **32.65**% | **24.49**% | **24.68**% | 0.077 | 0.000 | 0.025 | n/a |
| | MCTA | 12.99% | 26.53% | 4.08% | 15.58% | **0.132** | 0.000 | **0.036** | n/a |

existing topics. Then, the user can add words as anchors to the new topic. These features are similar to the ones used for interactively modeling monolingual topics [12].

Once the user finishes choosing anchor words for each topic, they press "Update Topics". This is a signal for MTAnchor to retrieve new anchor words from the interface and run multiword anchoring (Section 2.2). The algorithm approximates $\bar{Q}_w$ for every word $w$ in the vocabulary and then recomputes the topic matrices for each language. When MTAnchor finds new topics, the user can see the updated topics on the interface. At this point, anchors no longer have to be linked by dictionary entries because MTAnchor does not select anchors based on Equation 6. After the initial alignment, users define anchors and customize the topic model to their own needs.

## 4    Experiments

The first dataset consists of Wikipedia articles: 11,043 in English and 10,135 in Chinese. We shorten the articles to contain no more than three sections. We lemmatize the English articles using WordNet Lemmatizer [28] and segment the Chinese articles using Stanford CoreNLP [29]. For both languages, the articles fall under one of six categories: film, music, animals, politics, religion, and food.

Another dataset consists of Amazon reviews: 53,558 in English and 53,160 in Chinese (mostly from Taiwan) [30]. Each review has a rating, ranging from one to five. Since about half of the reviews have a rating of five, we change the classification task to a binary problem by labeling reviews with rating of five as "1" and the rest as "0". For the Wikipedia and Amazon datasets, the training-test split is set to 80:20. For the Chinese-English dictionary, we use entries from MDBG.[3]

To test low-resource languages, we use data from the LORELEI Sinhalese language pack [31]. These language packs are created to develop technologies that can process data in low-resource languages. In the pack, only a small subset of documents are labeled based on need type.[4] So, we treat the classification task as a semi-supervised problem. There are eight possible labels: evacuation, food supply, search/rescue, utilities, infrastructure, medical assistance, shelter, and water supply [32]. Out of the 1,100 (4,790) English (Sinhalese) documents, only 77 (49) of them have labels. For each language, half of the labeled documents are in the training set and the other half are in the test set. For the Sinhalese-English dictionary, we use entries from the LORELEI Sinhalese language pack.

We run experiments to evaluate three methods: multilingual anchoring, MTAnchor, and MCTA (Multilingual Cultural-common Topic Analysis) [33]. We choose MCTA as a baseline because it is a recent work on multilingual topic models with readily available code and aligns topics using a bilingual dictionary. We train models on multilingual anchoring and MCTA with twenty topics. For MTAnchor, we initially show users twenty topics, but the final number of topics is their choice. All methods are implemented in Python on a 2.3 GHz Intel Core i5 processor.

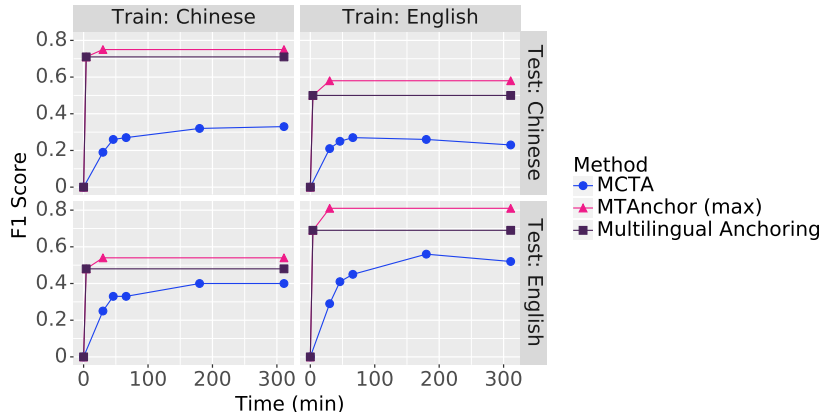

Figure 4: Classification accuracy over time until MCTA converges. For the Wikipedia dataset, multilingual anchoring converges within 5 minutes, but MCTA takes 5 hours and 18 minutes to converge. Multilingual anchoring outperforms MCTA in speed and classification accuracy.

The data for the MTAnchor user study are the English-Chinese Wikipedia articles. We invite twenty participants on Amazon Mechanical Turk (MTurk) to partake in the study. Each user is given thirty minutes to interact with the interface.[5] MTAnchor scales with the number of unique word types, rather than number of documents or number of words in the documents, so updates to the system take no longer than seven seconds on average. We only approve HITs from workers who have completed the task for the first time. After worker finishes the task, the interface provides a unique code for them to enter on MTurk. These rules ensure fair assessment of workers' interaction with MTAnchor.

## 4.1 Evaluating multilingual topics

Ideally, topic models should have topics that are *interpretable* and *useful* as classification features. So, we primarily base evaluation on two measures: classification accuracy and topic coherence. Measuring topic coherence considers both intrinsic and extrinsic scores [34]. The difference between the two is the reference corpus.[6] The intrinsic score uses the trained corpus itself, whereas the extrinsic score uses an external, larger dataset. The Sinhalese extrinsic coherence scores are not available because a large reference corpus cannot be formed for low-resource languages. By measuring both, we can evaluate the model's interpretability within a local and global context.

We evaluate these metrics separately for each language: English (EN), Chinese (ZH), and Sinhalese (SI). To classify labels from topics, we use the same procedure as described in Section 3.1. Then, we measure intra-lingual (I) and cross-lingual (C) accuracy with F1 scores. Intra-lingual accuracy refers to percentage of documents classified correctly using a classifier trained on documents in the *same* language. Cross-lingual accuracy refers to percentage of documents classified correctly using a classifier trained on documents in a *different* language (testing the algorithm's ability to generalize). For topic coherence, we use the NPMI (normalized pointwise mutual information) variant of automated topic intepretability scores over the fifteen most probable words in a topic [34]. For intrinsic scores (I), we use the trained corpus itself as the reference corpus. For extrinsic scores (E), we use 2.2M English Wikipedia articles and 1.1M Chinese Wikipedia articles.

During the user study, we hold out 100 documents as a development set for each corpus. Each time the user updates topics, the interface shows classification accuracy on the development set. When the user finally submits final anchor words, we evaluate their topics on the test set.

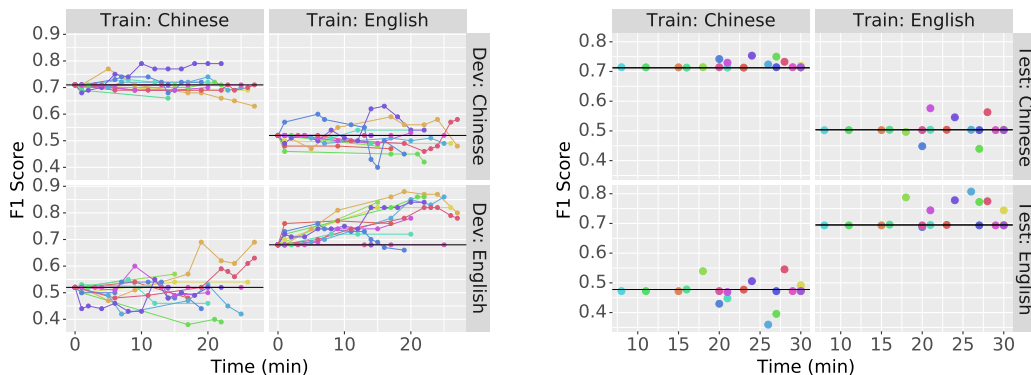

Figure 5: Classification accuracy of each participant in the MTAnchor user study over time. Each plot indicates the language of topics that the classifier is trained on and the language of topics that the classifier is tested on. The black horizontal line denotes multilingual anchoring score (no interactive updates). Each colored line represents a different user interaction and shows the fluctuation in scores on development set (left). Each colored point represents the final classification score on the test set; the point's x-coordinate indicates total duration of user's session (right).

## 4.2 Results

In experiments, multilingual anchoring converges much faster than MCTA (Figure 4). We compare scores across experiments for multilingual anchoring, MTAnchor, and MCTA, but only report the maximum and median scores from MTAnchor user experiments (Table 1). For English-Chinese datasets, multilingual anchoring performs better than MCTA in all metrics. For English-Sinhalese LORELEI dataset, topics from multilingual anchoring are more useful for classification tasks but are less coherent than MCTA topics.

In every metric, the MTAnchor maximum score across all users is higher than scores from other methods (Table 1). The MTAnchor median score across all users is approximately same as those of multilingual anchoring for all metrics. A few users outperform multilingual anchoring by spending more time interacting with the model (Figure 5). Within thirty minutes, a user can improve topic coherence and reach up to a 0.40 increase in any one of the classification scores.

## 5 Related work and discussion

Prior work on multilingual topic models mainly follow a generative approach. The Polylingual Topic Model [1] assumes that documents are topically aligned to track topic trends across languages. JointLDA [3] makes use of a bilingual dictionary and introduces "concepts" as a way to connect words from different languages. The model learns better monolingual models through optimizing cross-lingual corpora than LDA does when trained only on monolingual data. The Polylingual Tree-based Topic Model [7] builds tree priors to incorporate word correlation and document alignment information. MCTA [33] is another generative, multilingual model, but uses dictionary entries to capture "cultural-common" topics.

Multilingual anchoring is a spectral approach to modeling multilingual topics. The algorithm converges much faster than generative methods (Figure 4) and resulting topics form better vector representations for documents (Table 1). An advantage of anchoring over generative models is its robustness and practicality [14]. Generative methods need long documents to correctly estimate topic-word distributions, but anchoring handles documents of any size [13]. This is evident in models built on the Amazon dataset, which contains reviews with only one to three sentences. The health topic for multilingual anchoring is more interpretable than that of MCTA (Table 2).

Arora et al. [14] observe that more specific words appear in the top words of anchor-based topics. This is clearly shown in the LORELEI experiments; a topic from MCTA has general words like "help" and "need", while a topic from multilingual anchoring has specific words like "aranayanke" and "nbro" (Table 2). Both topics are about the 2016 Sri Lankan floods, but the topic from MCTA cannot

Table 2: Top seven words of sample English and Chinese topics are shown with anchors bolded. Topics from multilingual anchoring and MTAnchor are more relevant to document labels, thereby making them more useful as features for classification.

| Dataset | Method | Topic |
|---|---|---|
| Wikipedia | MCTA | dog san movie mexican fighter novel california |
| | | 主演 改編 本 小説 拍攝 角色 戰士 |
| | Multilingual anchoring | **adventure** daughter bob kong hong robert movie |
| | | 主演 改編 本片 飾演 **冒險** 講述 編劇 |
| | MTAnchor | **kong hong movie** office martial box reception |
| | | 主演 改編 飾演 本片 **演員 編劇** 講述 |
| Amazon | MCTA | woman food eat person baby god chapter |
| | | 來貨 頂頂 水 耳機 貨物 張傑 傑 同樣 |
| | Multilingual anchoring | eat diet food recipe **healthy** lose weight |
| | | **健康** 幫 吃 身體 全面 同事 中醫 |
| LORELEI | MCTA | help need floodrelief please families needed victim |
| | Multilingual anchoring | aranayake warning landslide site missing nbro areas |

specify the "need" type of documents. So, accuracy is higher when using topics from multilingual anchoring to classify documents. However, LORELEI experiments show that multilingual anchoring topics are less interpretable than MCTA topics. This might be caused by the obscure top topic words. Arayanake is a Sri Lankan town and "nbro" stands for National Building Research Organization. These words may have lowered coherence because they do not co-occur frequently with other top topic words. In this case, using MTAnchor can possibly increase topic coherence.

In the user study, a few participants create topics that are more applicable for specific tasks. In one experiment, a user finds the topic with anchor words "adventure" and "冒險(màoxiǎn)" too vague. The user knows that the task is to classify Wikipedia articles into one of six categories, so they add movie-related terms as anchors, like "movie", "演員(yǎnyuán)", and "編劇(biānjù)". Afterward, their topics significantly improves in classification accuracy and coherence. Other participants do not significantly change the topic model through interactive updates. More work can look into improving MTAnchor so that updates change topic distributions more drastically.

Interestingly, the scores for English topics increase considerably after user interaction compared to Chinese topics (Table 1). The participants are anonymous MTurk workers, so we are not aware of their language skills. We believe that workers are most likely fluent in English because the MTurk website is only available in English. If this fact holds true, then it can explain why the English topics have much higher scores than the Chinese ones. It also shows that people can improve topic models with prior knowledge, which supports the need for human-in-the-loop algorithms. In the future, it would be interesting to observe how language fluency affects quality of multilingual topics.

# 6 Conclusion

We present spectral and interactive topic models for multilingual document collections. The goal is to bridge the language gap using a multitude of resources: a dictionary, corpora, statistical models, and human input. A model that relies entirely on one resource is impractical for use in many settings, especially for low-resource situations. Multilingual anchoring can work with or without label supervision. Dictionary entries can be scarce or not fully accurate. People can use MTAnchor without a deep knowledge of topic modeling or machine learning. The method's versatility and speed make it an alternative to models like neural networks, which need a preponderance of labeled data. Future work can focus on understanding the effect of human input on multilingual topic models and accurately reflecting their feedback in cross-lingual representations.

**Acknowledgments**

We thank the anonymous reviewers for their insightful and constructive comments. Additionally, we thank Leah Findlater, Jeff Lund, Thang Nguyen, Shi Feng, Mozhi Zhang, Weiwei Yang, Eric Wallace, and Manasij Venkatesh for their helpful feedback. This work was supported in part by the JHU Human Language Technology Center of Excellence (HLTCOE) and Raytheon BBN Technologies, by DARPA award HR0011-15-C-0113. Any opinions, findings, conclusions, or recommendations expressed here are those of the authors and do not necessarily reflect the view of the sponsors.

## Footnotes

[1]Comparable corpora across languages are collections of documents about the same themes but that are *not* translations. Compared to more typical parallel data [15, 16], comparable data are more challenging.

[2]http://github.com/forest-snow/mtanchor_demo.

[3]https://www.mdbg.net/chinese/dictionary?page=cc-cedict.

[4]Documents in LORELEI language pack have multiple need types, but we have simplified the classification task by assigning only the first label to each document.

[5]Synopsis of user instructions: "There are 11,000 English Wikipedia articles and 10,000 Chinese Wikipedia articles, which belong to one of six categories: film, music, animals, politics, religion, food. Your goal is to find topics that can help classify documents within 30 minutes."

[6]Measuring topic coherence requires a reference corpus to sample lexical probabilities.

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
