[Reviews · NeurIPS 2018]

Reviewer 1



The paper proposes an anchor-based approach for learning a topic model in multilingual settings. The model first is built automatically and then can be refined by interaction with a user. The paper addresses a problem of interest of the previous NIPS submissions. Quality: The paper is technically sound. I like that for bilingual topic modeling it is only require to have a dictionary between two languages but documents should not be aligned for training a topic model. The interactive part could be interesting for applications but is not quite impressive from the methodological point of view since building a model interacting with a user is based on existing methods where the proposed approach for multilingual anchoring is used only for initial topic learning. The main missing piece in the paper from the technical point of view is absence of description of main optimisation algorithm. How exactly the authors solve their equation (5) in an efficient way. Could the authors please explain how do they choose the number of anchoring words if they allow multiword anchoring? Or are multi-words only used in the interactive settings? It also would be beneficial to see more discussion about conceptual comparison of generative and anchor-based approaches for topic modeling. Other suggestions/questions: 1. It would be an interesting comparison if MCTA results are provided both for limited and unlimited time scenarios. Now it is unclear whether MCTA gives poor results because it has not converged or because it generally learns poorer topics. 2. Follow-up regarding comparison with MCTA, the authors discuss briefly computational time of the methods without specifying what programming languages and hardware are used for implementations 3. Does the method have to be bilingual? It seems that it can be extended for more languages. If it is so, it could be mentioned in the paper, if the method is limited to be bilingual, it should be clearly stated. 4. I am not sure about user study experiment setup because the users were asked to essentially optimise classification performance of the system rather than interpretability of the topics directly. Interpretability is something that they probably based their changes on but this was not their primal goal. And it could be a case that a user made a change to improve interpretability of topics but this harmed classification performance since classification performance is not a direct measure of interpretability of topics and may not be related to it. As follow-up, the arguments in lines 255-257 about why one should care about interpretable features it is again talked about as classification is the end goal of topic modeling which is not correct since then one should most probably use some deep learning approaches rather than topic modeling 5. Figure 3b. What is the baseline here? Is it MCTA? Or is it multilingual anchoring? Clarity: The paper is mostly well written. I like a lot of examples used, this is really engage reading. Below are some suggestions for improvement: 1. Figure 2 is unreadable. The plots should be enlarged 2. Figure 3b. It is unclear why the plot type has changed from the one used in Figure 3a. There are lines in Figure 3a, but scatter points in Figure 3b. 3. At the beginning of Section 4.1 it is better to remind a reader that “multilingual anchoring” is the proposed method without human interaction and “MTAnchor” is the proposed method with human interaction 4. Table 1 is missing an actual caption what this table contains. The current caption discusses the results 5. Lines 234-236 specifying the evaluation procedure should be before user study experiment since it uses the same procedure 6. It is better to specify that NPMI is 7. Section 4.1 misses description of results from Table 1. For example, the authors could move the current caption of Table 1 into the main body of the paper. 8. Line 245: “a user can reach at least a 0.40 increase” – since the median results among users are the same as multilingual anchoring should it be “a user can reach UP TO a 0.40 increase”? 9. Agruments in line 255-257 10. Section 6 on Related work should probably be moved before Section 3 as in Section 3 the authors refer to JointLDA and in Sections 4 and 5 the authors compare their method with MCTA, both of which are introduced in Section 6. 11. Last sentence of the paper does not seem to be of any use – a very vague statement. Minor things: 1. S_k in equation (3) is not defined 2. The sentence “A computationally attractive …” in line 43 appears quite rapidly after general words about topic modeling. Maybe some more introductory words could help to direct the subject towards an anchor word approach 3. Typo: inperfection -> imperfection (line 113) 4. The footnote 3 is better to be stated in the main body of the paper 5. Typo: line 161 “not enough translations may be unavailable”. Should it be “available”? 6. Typo: line 192 “anchors are defined anchors so that they take full control …”. Should it be “users define anchors …”? 7. Typo: line 264 missing articles before “experiments” and “multilingual anchoring algorithm” 8. Typo: line 266 grammatically incorrect sentence. Should it be “approach to modeling cross-lingual topics THAT is both fast and accurate”? 9. Typo: line 267 “between topics if not training time is not long enough” – should first “not” be removed? Originality: The paper addresses an interesting problem in machine learning. The multilingual anchoring algorithm appears to be fairly novel while interactive MTAnchor seems to be a good contribution only as an application rather than a novel method. The related work is cited well for generative multilingual models and anchor-based methods, while alternative approaches for multilingual data processing are not discussed such as, e.g., Xiao and Guo, 2013 “A Novel Two-Step Method for Cross Language Representation Learning” Significance: The paper extends the anchor-based approach for topic modeling for multilingual data and could be a nice contribution in that area. The empirical results on comparison to one generative topic model show superiority of the proposed method, although as mentioned above it is unclear whether it is a matter of restricted time for training or general inability of generative topic model. In any way, the proposed method would have an advantage of faster convergence. As mentioned above, the interactive method proposed in the paper does not seem to provide a significant contribution. UPDATE AFTER RESPONSE: I would like to thank the authors for their response and addressing many of the concerns raised by the reviewers. Based on the response I would recommend the authors to focus more in general on a topic modeling (exploratory) point of view and less on classification and use classification accuracy also as a way to show how good is a found topic model, as an indicator of interpretability of topics. Also I would strongly encourage the authors to provide the results of MCTA after its convergence (maybe together with unconverged results) and either claim that the proposed algorithm both is more efficient and finds better topics than MCTA or that it is only more efficient if the results are competitive. The current version with only unconverged MCTA results raises questions and it is not very convincing.

Reviewer 2



The author(s) present a multilingual topic modeling approach based on anchor word based topic models with provable guarantees (see Arora et al 2013). The author(s) make two main contributions, first, they propose a greedy approach to find multilingual anchor words to match topics over different languages. As a second step, the author(s) study the effect of using humans to interactively improve the model using a human-in-the-loop approach (see Hu et al 2014). The paper is good and interesting. The experiments are reasonable and are well motivated. This is also the case for the proposed approach to identify multi-language anchor words, an approach that seems to beat previous approaches to dictionary-based multilingual topic models. The paper is clearly written and easy to follow. The only question is how big the contribution is. Some minor issues: a) The author(s) motivate decision and the use of anchor words for increased speed. At multiple places, this is motivated by natural disasters where time is of the essence (see line 2-3, 23-24, and 214-215). I agree with the author(s) that speed is important, especially in an interactive situation, but I can’t see the connections to disasters or similar situations. Nothing in the experimental evaluation connects to this situation (where Wikipedia and Amazon are analyzed). I think the author(s) need to motivate this with a reference where this kinds of models actually have been used in those situations or remove/tone down that argument in the paper. b) At line 10-11 the author(s) argues that the approach is used to correct errors. This is not shown in the experiments in any way, what is shown is rather that the users can improve the models. c) At line 26-27 the authors propose the anchor words as a way of incorporating knowledge into models. The paper cited (10) focus, to my knowledge, on anchor word for the purpose of separability. Especially since the identifying the anchor words is done using a greedy algorithm rather than using human knowledge. But I may be wrong. d) At line 123: I think the definition of d should be in the text, not as a footnote. Without a definition of d, equation 4 does not make any sense. e) One of the main evaluation metrics used in the paper is the classification accuracy for a 6 label classification task using an SVD classifier (see for example line 226-229). This evaluation metric will actually measure how well the model can learn topics that can represent these labels. This is not the same thing as how good the model is. I think it would be good to have perplexity or some similar measure that explain how well the different model fit the data. f) The issue of classification accuracy also affect the experimentation results with the 20 users. It is not clear from the paper if the users knew the 6 labels that they were classifying. If they did know, that would be a different task than if the didn’t know, so I think this needs to be stated. I think this is partly mentioned by the discussion on "adventure" at line 258-261. g) One issue with the experimental result is that it is not clear how often the MCTA topic model did not converge. The author(s) used models that had not converge for 30 minutes. That make it difficult to assess how good the baseline actually is. I think it would be good to know how long it would have taken for a MCTA model to converge for the given data to assess how fair the comparisons are. h) The sentence “We could train a neural network to predict labels for Chinese documents” at line 256 is a little difficult to understand. It is not obvious what the author(s) want to say. Typos: i) In equation 3 I guess it should be \mathcal(G) under the summation sign. Also S_k is not defined anywhere. ii) At line 113, “inperfection” should maybe be “imperfection”. iii) Reference 4 is missing the publication year in the bibliography. Quality: The paper is in general of good quality with the experiments showing positive results. The main questions concern issue e, f and g above, that concern the choice of classification accuracy as the main evaluation metric and how the time limit of 30 minutes for the MCTA model affect the experimental results. Clarity: The paper is very well written. It is clear what the authors do, the approach/method and how the experiments has been conducted. There are some minor issues that can further improve the clarity of the paper. Originality: I may have missed some reference, but I do think the idea of multilingual anchor words are new. The basic ideas are built upon known techniques (anchor word topic models and interactive topic models). I generally think that the originality is the weakest part of the paper, since the additional contribution in the form of identifying multilingual anchor word is not a very large contribution as such, even though it is nice. Significance: The significance of the paper is a little hard to judge. The paper is well written and the results from the experiments are good. The idea of multilingual anchor-words are also straightforward to use so that approach may very well be used in further work using topic models with provable guarantees. UPDATE AFTER REBUTTAL: I think the rebuttal clarified some of the issues with the paper. But I'm still critical with the focus on classification accuracy. In the rebuttal, the authors clearly state that this is the goal: "However, ideally we want topics that are interpretable and can improve classification accuracy." and "Our goal is to create interpretable topics that can also act as additional features to improve accuracy in classification tasks.". But at the end of the rebuttal, the authors conclude that classification is not the best approach to for topic model usage: "We want to clarify that MTAnchor is not a tool solely for finding features that can optimize classification performance. In that case, a neural network should be used." So this makes it a little confusing what the purpose of the paper is. Even though I think the basic idea of multilingual anchor-words is good (as an example to explore multilingual corpora), the focus on classification as the goal weakens the paper. Also, the criticism of the argument for disaster relief is not discussed in the rebuttal, although raised by me and reviewer 3. This affects the question regarding the convergence of the MCTA. If it just takes 4-6 h to get the model to converge, the question is how well a converged model would behave as a baseline. Without knowing this in Table 1, I'm not sure Table 1 is a fair comparison, especially if there is no motivation why the models only have a computational budget of 30 minutes. Finally, the authors point to Chang et al (2009) as an argument for not using perplexity. But perplexity still shows how well the model fits the data, even though this is done by the model in a, from a human perspective, incoherent way. So including perplexity (or similar measures) would strengthen the paper even though perplexity won't say how coherent the model is.

Reviewer 3



This work presents an extension of the anchor words topic modeling algorithm to the multilingual setting through two pieces: a greedy strategy to automatically select multilingual topic anchors that optimally expand the subspace of both languages, and an interactive workflow for adding and modifying anchor words for topics. Both models exceed the existing state of the art in interactive topic modeling, with the interactive workflow in particular performing the best. I really like the balance of this work between a new algorithm, a new workflow, and a variety of evaluations; the work reads clearly and contributes meaningfully to the topic modeling literature. I think one piece I was missing was the clear motivation of the interactive multilingual modeling case; while the example of disaster relief message triage was invoked repeatedly, it was not totally clear what the intended use of a topic model instead of disaster-related keywords was in that application. It seems as if the process is focused on optimizing some known development task, e.g. classifying a labeled subset of the multilingual data and using that classification to extend to unseen text, but it is still not totally clear to me that topic models are the right approach to that. That is a lot more to approach for this paper than is necessary, but it seems worth addressing a bit more head-on given the running example. I thought some of the writing on the interactive workflow could have used some additional development. Part of this may be because the figures displaying the interactive portions were a bit hard to read, as they are exceedingly small; making sure there is enough space for them in the camera-ready seems important for the model being presented. It also skimps a bit on details on what happens with topic addition and deletion - how are new words selected if a topic is deleted? Are other anchors downweighted if one close to them is rejected? Some additional references to the workflows for topic addition and removal from the referenced papers [9] and [12] in section 3.2 might help with filling in that quickly. A recurring problem mentioned in the anchor word literature that motivates a lot of the development of new anchor-finding algorithms is that the "best" anchors in terms of their independence from other vectors and placement in a convex hull tend to be very rare, almost obscure words. Tandem anchors seem to help with this problem a bit, using a combination of words to get the individual meaning, but it seems like the interplay between that problem and the multilingual setting (where rare words are less likely to be translated) is interesting and worth discussing. Does the multilingual case actually help ensure these words are more common? I had one more mathematical concern about the weighted minimum of distances in the two corpora in equation (5): are these metrics balanced? I know they are looking at entries in a probability matrix, but I would expect that corpora with different-sized vocabularies might have a tendency towards one language consistently having the minimum distance over the hull. I was wondering if there was a reason no normalization was necessary, either through choice of distance metric or other observation about the data. I had some smaller notes about the writing: - line 54, "may be explained by" is a bit ambiguous - may co-occur with/have components represented by? - line 59, "the probability that a token of word type i belongs to topic k" is probably clearer? - using anchors as an example was clever and made me smile, a lot - line 192, "anchors are defined anchors" is this supposed to say "authors define anchors" or "anchors are defined by authors"? - table 1, bolding the MTAnchor (max) result would probably be okay - it seems worth having a few words about how you selected your MTurk workers to ensure your system wasn't cheated - line 264, I'm not sure that "supersedes" is actually the word you want there - it again might be good to revisit papers 9 and 12 in the related work, since they are close on the interactive work Ultimately, I enjoyed this paper a lot, and believe it would be a worthy contribution to the NIPS program. AUTHOR RESPONSE: I appreciate the author's notes in addressing some questions. In light of the information about model convergence being a little slower than I had initially thought, I would encourage the authors to spend some time reconsidering the choice of disaster response as their primary motivating example in favor of something less extremely time-sensitive. An example would be identifying news stories or Wikipedia articles in one language about a subject not being represented in another language.